# ADDRESS2VEC: GENERATING VECTOR EMBEDDINGS FOR BLOCKCHAIN ANALYTICS

## ABSTRACT

Bitcoin is a virtual coinage system that enables users to trade virtually free of a central trusted authority. All transactions on the Bitcoin blockchain are publicly available for viewing, yet as Bitcoin is built mainly for security its original structure does not allow for direct analysis of address transactions. Existing analysis methods of the Bitcoin blockchain can be complicated, computationally expensive or inaccurate. We propose a computationally efficient model to analyze bitcoin blockchain addresses and allow for their use with existing machine learning algorithms. We compare our approach against Multi Level Sequence Learners (MLSLs), one of the best performing models on bitcoin address data.

## 1 INTRODUCTION

Bitcoin(Nakamoto) is a virtual coinage system that functions much like a standard currency, enabling users to provide virtual payment for goods and services free of a central trusted authority. Bitcoin relies on the transmission of digital information, utilizing cryptographic methods to ensure secure, unique transactions. Individuals and businesses transact with the coin electronically on a peer-to-peer network utilizing a shared transaction ledger (the Blockchain). It caught wide attention beginning in 2011, and various altcoins a general name for all other cryptocurrencies post-Bitcoin soon appeared

It has placed itself as the most widespread and commonly used cryptocurrency with no signs of slowing down(Chan et al., 2017). Representing over 81% of the total market of cryptocurrencies(coi), Its market capitalization is estimated to be approximately $177.8 Billioncoi accounting for about 90% of the total market capitalization of Virtual Currencies(Houben & Snyers). Bitcoin uses public key cryptography to generate secure addresses for users where each address is a public key, and use of the bitcoins stored in it requires signing with a private key. These address identifiers are used by their owners to hold bitcoin pseudonymously. A typical Bitcoin transaction consists of two sets: a set of source addresses and a set of destination addresses. Coins in the source addresses are collected and then sent in differing amounts to the destination addresses. (Houben & Snyers)

While bitcoin address data is publicly available, it is not straightforward to analyze address transaction data since it is not aggregated in one block/place.

## 2 CURRENT METHODS

We are comparing our work to multi level sequence learners (MLSLs)(Agrawal et al., 2016) based on the use of multiple levels of LSTMs(Hochreiter & Schmidhuber, 1997) to generate a tree of depth $D$ defined by $0 < d \leq D$ for each node where each layer $d$ represents the dth neighbourhood of the node. The maximum depth of the tree $D$ is arbitrary and picked to suit accuracy and resources. Multi-level Sequence Learners are a class of Long Short Term Memory Networks optimized for generating sparse embeddings of graphs LSTMs scale with depth as for each child node there is at least a learner. However, we develop address2vec as a computationally cheaper and comparable approach extended from previous embedding generation models(Grover & Leskovec, 2016)(Mikolov et al., 2013).

## 3   METHODOLOGY

We extract transaction data for each block from a range of selected blocks, we developed a python script[1] that connects to blockchain.info(blo) and queries for block data. We then use an autoencoder that compresses the features of a transaction(Time, Block Height, Size, Input address count, output address count, input value, output value, number of outputs not part of a change transaction, transaction fee) of a transaction into a single number. For each block we extract transactions and construct a transaction graph $G = (A, T)$ using networkX(Hagberg et al., 2008) where $A$ is the set of addresses $a$ on the blockchhain where and $T$ is the set of transactions on the blockchain. Where each transaction $t_{a_s a_r} := (a_s, a_r, w_{btc})$ has a sending address $a_s$ and a receiving address $a_r$ and the autoencoder compressed features $w_{btc}$. We use the same weight $w_{btc}$ for all input output pairs in a single transaction. We then proceed to use the node2vec algorithm(Leskovec & Sosič, 2016)(Grover & Leskovec, 2016) to generate node embeddings for the given graph. We then use a small densely connected network to predict behavior of addresses. We predict whether or not a bitcoin address will be empty after 1 year and compare our results to MLSLs and a weighted coin toss baseline.

## 4   RESULTS

We are able to generate comparable results to MLSLs at a cheaper computational cost. Refer to Table 1 below.

Table 1: Node Classification Prediction Results

|  | Avg. Recall | F-1 (Spent) | F-1 (Hoard) |
| --- | --- | --- | --- |
| Baseline | 0.50 | 0.79 | 0.20 |
| 1-MLSL | 0.75 | 0.82 | 0.62 |
| 2-MLSL | 0.78 | 0.85 | 0.64 |
| 3-MLSL | 0.77 | 0.84 | 0.63 |
| A2V (Dense FNN) | 0.71 | 0.75 | 0.45 |

## 5   CONCLUSION

It is apparent that address2vec is a significant improvement over a baseline approach, although not as accurate as MLSLs we believe further tuning of the model's architecture can yield a more accurate iteration of address2vec, especially making the model end to end differentiable, we currently use separate phases. We also plan to test address2vec on different bitcoin behavior tasks, measuring the similarity of various users and their relationships by measuring their vector distances and predicting market rates of bitcoin through analyzing most recent addresses on the blockchain.

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

# A    APPENDIX

You may include other additional sections here.

