# OpenReview forum: "Address2vec: Generating vector embeddings for blockchain analytics"
_ICLR.cc/2020/Conference — Reject_

### Official Review · AnonReviewer1 · 2019-10-09
**Official Blind Review #1**

**Rating:** 1

**Review:**

The paper proposes to use an autoencoder, networkX, and node2Vec in succession to convert a Bitcoin transaction to a vector. This is then used to predict whether a Bitcoin address will become empty after a year. The results are better than flipping a coin, but worse than an existing baseline.

Given the apparent lack of any technical contribution to machine learning theory or practice, the inconclusive empirical results, and the generally unpolished writing (e.g., long run-on sentence in the conclusion, vague problem definition), I do not believe this paper is suitable for publication.

**Experience Assessment:**

I do not know much about this area.

**Review Assessment: Checking Correctness Of Derivations And Theory:**

I carefully checked the derivations and theory.

**Review Assessment: Checking Correctness Of Experiments:**

I carefully checked the experiments.

**Review Assessment: Thoroughness In Paper Reading:**

I read the paper thoroughly.

---

### Official Review · AnonReviewer2 · 2019-10-16
**Official Blind Review #2**

**Rating:** 1

**Review:**

In this paper, the authors propose a new method for generating vector embeddings. The studied problem is important and the topic is related to area of Bitcoin.

Empirical studies on some dataset (no description about the dataset) show some results on some evaluation metrics (no clear description about the metrics). For the methods 1-MLSL, 2-MLSL and 3-MLSL, it is seems that they are better than the proposed one (i.e., A2V) on some metrics, which is then inconsistent with the claims in the paper.

My major concern is that the paper is a bit too short and is lack of some necessary information, for example:

1 The authors are encouraged to provide sufficient background introduction, so that the reader can have a big picture of the problem and area.

2 The authors are encouraged to provide a detailed discussion and justification about the motivation, as well as the challenges and intuitions about the proposed method in the Introduction Section.

3 The authors are encouraged to show a detailed derivation about the technical details, in particular its difference compared with the major baseline, i.e., MLSL.

4 The authors are encouraged to follow the typical writing about the experiments in a paper, e.g., description about the datasets, evaluation metrics, baseline methods, parameter configurations and results analysis, etc.

Based on the above comments, I have to make a reject recommendation.

**Experience Assessment:**

I do not know much about this area.

**Review Assessment: Checking Correctness Of Derivations And Theory:**

N/A

**Review Assessment: Checking Correctness Of Experiments:**

N/A

**Review Assessment: Thoroughness In Paper Reading:**

I read the paper at least twice and used my best judgement in assessing the paper.

---

### Official Review · AnonReviewer3 · 2019-10-27
**Official Blind Review #3**

**Rating:** 1

**Review:**

Authors propose to apply the existing machine learning model to analyze bitcoin blockchain addresses. It uses autoencoder to extract the feature of the transaction and construct a transaction graph. Then the node2vec algorithm is used to generate node embeddings for the given graph. The task is to predict the behavior of addresses.  The experiments are conducted against Multi Level Sequence Learners (MLSLs), one of the best performing models on bitcoin address data.

Pros:
This work studies an interesting and challenging problem.

Cons
1. This is an unfinished work. The proposed method lack of detail description.
2. The performance is much lower than the MLSLs baseline methods.

**Experience Assessment:**

I have published in this field for several years.

**Review Assessment: Checking Correctness Of Derivations And Theory:**

I carefully checked the derivations and theory.

**Review Assessment: Checking Correctness Of Experiments:**

I carefully checked the experiments.

**Review Assessment: Thoroughness In Paper Reading:**

I read the paper thoroughly.

---

### Decision · Program_Chairs · 2019-12-19

**Decision:**

Reject

**Comment:**

The paper propose to analyze bitcoin addresses using graph embeddings. The reviewers found that the paper was too incomplete for publication. Important information such as a description of datasets and metrics was omitted.